# Health Literacy and Nutrition of Adolescent Patients with Inflammatory Bowel Disease

**DOI:** 10.3390/nu17152458

**Published:** 2025-07-28

**Authors:** Hajnalka Krisztina Pintér, Viola Anna Nagy, Éva Csajbókné Csobod, Áron Cseh, Nóra Judit Béres, Bence Prehoda, Antal Dezsőfi-Gottl, Dániel Sándor Veres, Erzsébet Pálfi

**Affiliations:** 1School of PhD Studies, Semmelweis University, 1085 Budapest, Hungary; 2Faculty of Health Sciences, Department of Dietetics and Nutritional Sciences, Semmelweis University, 1085 Budapest, Hungary; nagyviolaanna@gmail.com (V.A.N.); csajbokne.csobod.eva@semmelweis.hu (É.C.C.); palfi.erzsebet@semmelweis.hu (E.P.); 3Pediatric Center, Semmelweis University, 1085 Budapest, Hungary; cseh.aron@semmelweis.hu (Á.C.); beres.nora@semmelweis.hu (N.J.B.); prehoda.bence1@semmelweis.hu (B.P.); dezsofi.antal@semmelweis.hu (A.D.-G.); 4Department of Biophysics and Radiation Biology, Semmelweis University, 1085 Budapest, Hungary; veres.daniel@semmelweis.hu

**Keywords:** health literacy, nutrition, inflammatory bowel disease, adolescent, Mediterranean diet

## Abstract

**Background/Objectives**: Nutrition in inflammatory bowel disease (IBD) is a central concern for both patients and healthcare professionals, as it plays a key role not only in daily life but also in disease outcomes. The Mediterranean diet represents a healthy dietary pattern that may be suitable in many cases of IBD. Among other factors, health literacy (HL) influences patients’ dietary habits and their ability to follow nutritional recommendations. The aim of this study was to assess HL and dietary patterns in adolescent and pediatric patients with IBD. **Methods**: We conducted a cross-sectional study that included a total of 99 participants (36 patients with IBD receiving biological therapy recruited from a single center and 63 healthy controls). HL was assessed using the Newest Vital Sign (NVS) tool regardless of disease activity, whereas diet quality was evaluated by the KIDMED questionnaire exclusively in patients in remission. Linear regression models were used to evaluate the effects of sex, age and group (patients vs. control) on NVS and KIDMED scores. **Results**: Most participants (87.9%) had an adequate HL, which was positively associated with age. While the most harmful dietary habits (such as frequent fast-food consumption) were largely absent in the patient group, KIDMED scores indicated an overall poor diet quality. **Conclusions**: Although HL increased with age and was generally adequate in this cohort, it did not translate into healthier dietary patterns as measured by the KIDMED score. Further research with larger, more diverse samples is needed to clarify the relationship between HL and dietary adherence in adolescents with IBD.

## 1. Introduction

Inflammatory bowel disease (IBD) profoundly affects patients’ nutrition; however no universally recommended diet exists that would suit all patients. The disease subtype, the patients’ current state, personal preferences and intolerances must be considered when making a diet recommendation. Based on the guidelines [1] the exclusive enteral nutrition (EEN) is one of the most important tools for the induction of remission in pediatric Crohn’s disease (CD), lately research focuses on more tolerable and palatable diets such as partial enteral nutrition (PEN), Crohn’s disease exclusion diet (CDED), the specific carbohydrate diet (SCD) [2,3,4] as possible ways towards the induction or maintenance of remission. In accordance with the latest ESPEN guideline, the CD exclusion diet in combination with partial enteral nutrition should be considered in certain cases [5]. Furthermore, a healthy dietary pattern (with avoidance of individual nutritional triggers) should be followed in the remission phase. The Mediterranean diet (MD) is also one of those diets, that seem to improve patients’ quality of life [6,7,8,9] and has a positive impact on the disease course [10]. It may be recommended for patients with IBD in the absence of contraindications [11].

Adherence to the Mediterranean diet in IBD is associated with intestinal inflammation [12]. Furthermore, disease activity, surgical history, and quality of life also affect adherence [13,14]. Several tools are available to measure dietary adherence to MD; however the KIDMED (Mediterranean Diet Quality Index) [15] questionnaire has been specifically validated for children and adolescents, and many researchers applied it with pediatric IBD patients earlier [12,16,17]. It consists of 16 questions and categorizes the results as poor, average and good adherence to MD.

Beyond dietary patterns, health literacy (HL) is among the various factors that may affect disease outcomes in IBD. Health and nutritional literacy are getting more interest in understanding food and dietary choices in patients and healthy individuals. It is clear that the level of health literacy determines dietary adherence and nutritional behaviors and evidence suggests that it also predicts the adherence to nutritional recommendations [18].

Research on health literacy shows that higher health literacy is associated with more favorable dietary choices such as consuming more fruits and vegetables [19]. Based on a comparative study, there are substantial differences between the levels of adolescents’ HL in European countries. The proportion of low HL varies from 6.0% to 17.7% and the proportion of high HL from 12.8% to 38% [20].

Evidence suggests that HL predicts the adherence to dietary behavior in individuals with nutrition-related chronic conditions [21]. However the negative effects of limited health literacy are non-negotiable, it is still an underexplored issue among children and adolescents particularly within the context of pediatric IBD [22,23]. Studies highlight that this population’s HL is low and needs to be improved [24,25].

Health literacy can also be assessed using a range of tools. The Newest Vital Sign (NVS) tool, among them, is a quick, interviewer-administered HL screening instrument with 6 questions about a nutrition label of an ice cream box. As a result, it classifies the HL of individuals in three levels: high likelihood of limited HL, possibility of limited HL and adequate HL.

The aim of this study was to investigate the health literacy and nutrition of adolescent patients with inflammatory bowel disease.

Our findings indicate that health literacy among adolescents with inflammatory bowel disease was generally adequate, as measured by the Newest Vital Sign questionnaire, with performance improving with age. However, their adherence to the Mediterranean diet, assessed using the KIDMED questionnaire, was predominantly poor.

## 2. Materials and Methods

### 2.1. Study Design and Participants

This study was conducted on IBD patients receiving biological therapy at the Bókay Street Department, Pediatric Center, Semmelweis University, Budapest and on healthy controls of a high school and a primary school in the capital (Budapest, Hungary) and metropolitan area of the capital. All subjects (over 18 years old) or their caregiver signed a declaration of consent.

### 2.2. Data Collection Period and Ethics Statement

The research was carried out from February 2022 until August 2024 with the ethical approval by Semmelweis University under the ethical approval code ETT-RKEB: 162/2020.

### 2.3. Methods

For the assessment of health literacy, the Newest Vital Sign (NVS) tool was used [26] and for the perception of diet, the KIDMED score [15] was applied in IBD patients in Hungarian versions.

The NVS is a widely utilized tool in clinical practice and has also been used by other researchers in both the age group and patient population examined in our study [27,28,29,30]. This questionnaire was chosen in addition to the above due to its availability of a validated Hungarian version [26].

Currently there is no dietary adherence questionnaire available for the official Hungarian food-based dietary guideline, but several ones were published and validated for the Mediterranean diet which is also known for modeling a healthy dietary pattern and reducing inflammation. We chose the KIDMED questionnaire because it fits our targeted population and is easy to execute in a clinical setting.

### 2.4. Participants’ Characteristics

A total of 99 participants were included in the study, comprising 36 patients (28 had CD, 7 UC and 1 unclassified IBD) and 63 controls. The predominance of CD patients is due to the more frequent use of biological therapy in CD. 24 males (66.7%) and 12 females (33.3%) were in the patient group. The KIDMED questionnaire was administered only to patients in remission to avoid distortion of the results due to dietary changes during disease flare-ups. Mean age in the patient group was 15.7 ± 2.35 years, the youngest patient was 10 and the oldest 19 years old. The control group consisted of 12 (13 children) and 16 (50 children) years old healthy individuals, resulting in a mean age of 15.4 ± 1.93 years.

### 2.5. Statistical Analysis

After data preparations we described the collected data on tables and plots.

To assess the effect of sex and group (patient or control) on the NVS and KIDMED scores, linear regression models were used adjusting for the effect of age. Two-level interactions were assessed, but if these were neither significant (at 5%) nor relevant, we excluded the interactions from the reported final models. We used diagnostic plots to assess the assumptions of the models.

The statistical analyses were created by using R software [31] (v4.4.1). Descriptive tables were generated using the table1 package [32] (v1.4.3), upset plots were crated with UpSetR [33] (v1.4.0) and other plots were produced with ggplot2 [34] (v3.5.1). For the interpretation of regression results, the gtsummary [35] (v1.7.2) and sjPlot [36] (v2.8.16) packages were applied.

We used 5% as significance level.

## 3. Results

### 3.1. Results of Health Literacy Screening

The Newest Vital Sign questionnaire consists of 6 questions, and it is scored by the correct answers. 0–1 points indicate a high likelihood of limited HL; 2–3 points suggest a possibility of limited HL and 4–6 points almost always refer to adequate HL.

Table 1 summarizes the NVS score outcomes for the control and patient groups. 91 participants (28 patients and 63 controls) completed the NVS questionnaire, and their mean score was 5.04 ± 1.47; only 4.40% had a high likelihood of limited HL, 7.69% a possibility of limited HL and 87.91% an adequate HL (8 patients did not complete this questionnaire). The mean score was slightly lower in the patient group (4.86 ± 1.18) vs. in the control group (5.13 ± 1.58); however, this difference was not statistically significant (Mann-Whitney U = 660.5, z = 1.900, *p* = 0.057). No one had a high likelihood of limited HL in the patient group. The minimum score in the patient group was 2, while it was 0 in the control group, which explains the 6.3% of high likelihood of limited HL found among controls. The maximum score was 6 in both groups, while the median was 6 in the controls and 5 in the patients.

Sex data were not available for the control group; however, among patients, the mean NVS score was higher in females (5.22) compared to males (4.68). It should be noted that this difference did not reach statistical significance based on the Mann-Whitney U test (U = 61; the critical value of U at *p* < 0.05 is 45). Accordingly, the result was not significant at the *p* < 0.05 level (z = −1.181, *p* = 0.238). Figure 1 shows the patients’ NVS scores by sex.

Linear regression models were used to assess the assumptions of the models. Table 2 and Figure 2 represent that there is a significant difference in the NVS score examining the two groups, not accounting for age (β = 7.9, (higher NVS score in the patient group) 95% CI: 4.3–11 [score], *p* < 0.001).

In the control group each additional year of age is associated with an increase of 0.74 score in the NVS scale, which is also statistically significant (95% CI: 0.58–0.90 [score/years], *p* < 0.001).

The significant interaction indicates that the effect of age differs between the patient and control groups. Specifically, in the patient group, the effect of age is much smaller than in the control group (β = −0.54 [score/years], 95% CI: −0.77 to −0.31 [score/years], *p* < 0.001). It suggests that the positive influence of age on the outcome is diminished in the patient group.

Table 3 and Figure 3 represent the analysis of sex and age in relation to NVS score in the patient group.

In the patient group the predicted values by sex indicate that female adolescents tend to have higher NVS scores with ageing, averaging 0.79 points higher than males (β = −0.79 [score], 95% CI: −1.7 to 0.10 [score]), though this did not achieve conventional statistical significance.

The effect of age was statistically significant (*p* < 0.010); each additional year of age is associated with a 0.23-point increase in NVS score (β = 0.23 [score/years], 95% CI: 0.06 to 0.41 [score/years], *p* < 0.010). This suggests that older adolescents tend to have higher HL levels as measured by the NVS.

As the categorized NVS score did not vary in a meaningful amount, we could not make an analysis of it.

For the analysis of the results by disease type, the Mann-Whitney U test was performed; however, no significant difference was found in the NVS scores between CD and UC patients (U = 65, z = 0.028, *p* = 0.976).

In addition to the regression results, an upset plot (Figure 4) was created to visualize response patterns in the patient group (*n* = 28).

The dots in the figure indicate the specific combination of variables represented in the upper section. The frequency of each individual variable is displayed on the left, while the frequency of each specific combination is shown at the top.

The most frequent response pattern was that 11 patients answered all six questions of the NVS correctly (first column). Notably, all participants (n = 28) answered the fifth question correctly, which is a reading comprehension test (*“Pretend that you are allergic to the following substances: Penicillin, peanuts, latex gloves, and bee stings. Is it safe for you to eat this ice cream?”).* In contrast, the fourth question, which required percentage calculation *(“If you eat 2500 kilocalories in a day, what percentage of your daily value of calories will you be eating if you eat one serving?”*), was answered correctly by the fewest participants (n = 19). No specific combination of questions emerged as being most frequently answered correctly apart from these findings.

### 3.2. Results of Diet Adherence Screening

The KIDMED questionnaire consists of 16 items, each of which requires a binary response of either ‘Yes’ or ‘No’. Four of the questions refer to negative aspects (such as fast-food consumption) and 12 to positive (for example fruit and vegetable consumption) connotations about the Mediterranean Diet. The index of adherence to the MD is calculated as the sum of the answers and ranged from −4 to 12. Then individuals are categorized into 3 groups: poor adherence (≤3), average adherence (4–7) and good adherence (≥8) to MD.

This questionnaire was only administered in the patient group (n = 29). As inflammation and restriction diets severely impact a patient’s food choices, those in acute phase and/or following an exclusion diet (CDED) or exclusive enteral nutrition (EEN) were excluded (7 patients).

The mean adherence score was 2.66 ± 2.02. 17 patients (58.62%) demonstrated poor adherence to the MD, 12 (41.38%) showed average adherence, and none met the criteria for good adherence (Table 4). Although the mean score was higher among females (3.33 ± 1.22) compared to males (2.35 ± 2.25), the difference was not statistically significant (Mann-Whitney U = 60.5, z = −1.367, *p* = 0.171). Among female adolescents the minimum score was 2 and the median 3, among males the minimum was −2 and the median 2. The highest score recorded was 5 for females and 6 for males.

Figure 5 represents the KIDMED scores achieved by the individuals.

No statistically significant differences in KIDMED scores were identified using inferential statistical methods.

The Mann-Whitney U Test detected no statistically significant differences between the KIDMED score results of CD and UC patients (U = 62, z = 0.349, *p* = 0.726).

For the construction of the upset plot ‘Yes’ or ‘No’ answers were recoded to ‘Good’ or ‘Bad’ from the perspective of nutritional recommendations, with the plotted dots representing the ‘Good’ responses (Figure 6). This was necessary because the scoring of the questionnaire is somewhat more complex. In certain cases, the response is scored as 0 or 1 point, while in others the result is 0 or −1 depending on whether the answer is ‘Yes’ or ‘No’. For example, in the case of question 6 (“*Do you eat fast food more than once a week?*”), a ‘Yes’ response yields −1 point, whereas a ‘No’ response yields 0 points. For question 7 (“*Do you consume legumes at least three times a week?*”), a ‘Yes’ response is scored as 1 point, and a ‘No’ response as 0 points.

No distinct pattern of item combinations emerged from the responses to this questionnaire. This may be partly attributed to the structure of the KIDMED tool, which comprises 16 items—a relatively high number that allows for substantial variability. Only in two instances did two participants provide identical sets of responses. Nevertheless, it was possible to identify certain items that were most frequently answered in the same way across participants.

The questions regarding the frequency of fast-food consumption, breakfast choices, and frequency of pastry and sweets consumption were most frequently answered with ‘No’, which represents the favorable response for these questions. In contrast, items such as fish consumption, nut consumption and legume consumption received the fewest ‘Yes’ responses, which is considered suboptimal in terms of dietary recommendations.

We did not create a model for categorical KIDMED score as using the numerical score had higher power.

Summarizing the findings, outcomes of the NVS questionnaire indicate an adequate health literacy in the population involved in this study. Reading comprehension skills were found to be better than percentage numeracy and HL is expected to increase with age.

The results of the KIDMED questionnaire indicate a poor quality of the patients’ diet involved in this study. However, the most negative elements of a Western diet—such as fast-food consumption, confectionery consumption—appear to be largely absent from the participants’ diet, protective components like the consumption of fish, legumes and nuts are also not an integral part of their nutrition.

No significant differences were found between the CD and UC groups regarding either health literacy or dietary adherence. Moreover, no statistically significant associations were observed between dietary adherence and health literacy levels, possibly due to the small study population.

## 4. Discussion

Health and nutritional literacy are increasingly popular fields of research, but there is still a lack of comprehensive data within adolescent patients with IBD.

The uniqueness of our research lies in the fact that we investigated these rarely assessed parameters that influence quality of life and everyday life within a specific patient population. We used reliable and validated questionnaires tailored to our target age group and we were the first to assess health literacy and adherence to the Mediterranean diet in this specific population in Hungary.

In patients with IBD, most health literacy studies focus specifically on disease-related knowledge and consistently report low levels of health literacy. Limited HL has been linked to worse patient-reported outcomes, depressive symptoms, and poorer healthcare outcomes—including longer hospital stays, higher readmission and complication rates—and some estimates suggest that up to 24% of IBD patients have inadequate health literacy [22,24]. Similar trends are observed in general gastrointestinal cohorts, where lower health literacy predicts increased hospital length of stay, readmissions, and complications [37]. However, it is gratifying that HL seems to be appropriate with the result of 87% of adequate HL in our study, but it needs to be investigated more widely and in different patient and age groups.

We used diagnostic plots to assess the assumptions of the models. We found the assumptions fulfilled, although it was a little questionable for the relation of NVS score in the patient and control group. This was maybe caused by the fact that the control group only consisted of 12- and 16-year-old children. Therefore, this model results should be handled with limitations. Linear regression analysis of HL score revealed that patients had significantly higher scores compared to controls. Age was positively associated with the score in the control group. However, the interaction term indicated that this positive association was weaker in the patient group. This suggests that while the score tends to increase with age in controls, the age-related increase is substantially smaller in patients. Overall, these findings confirm that age is positively associated with NVS score, meaning older adolescents tend to have higher health literacy as measured by the NVS, which corresponds with others’ findings [38,39,40]. Similarly, as age increased, the HL score also increased proportionally. The observed difference by sex indicates a trend toward lower scores in males. However, this was statistically not significant, the point estimate indicates a relevant effect, therefore it may be worthwhile to conduct the study on a larger sample as well.

Limited data are available on adherence to MD in individuals with IBD, especially adolescents. Our findings show a poor adherence to MD which is lower than in previous studies from this field [12,13]. Fiorindi et al. reported that patients demonstrated good dietary adherence, as indicated by an average Medi-Lite score of 10.4 [13]. However, they similarly found no significant differences between the CD and UC groups, as our study. Dietary adherence appeared to be more strongly associated with disease activity, with patients with CD showing significantly higher adherence during remission. This aspect was not investigated in the present study, as it was assumed a priori that patients would not follow a balanced diet during periods of disease flare. Based on Strisciuglio’s findings, who utilized the same dietary adherence measurement tool as our research, children with IBD had an intermediate adherence to MD [12], and they also were able to detect a correlation about MD and decreased intestinal inflammation. Cadoni et al. reported about a low adherence to MD among IBD patients, similarly to our study, as measured by the Medi-Lite score [14].

Our observation of low dietary adherence among patients in remission may be due to several underlying factors, the examination of which is recommended. It is possible that patients in remission are more prone to adopting unfavorable dietary patterns, as the lack of gastrointestinal symptoms may reduce their motivation to adhere to a healthy diet. Notably, analysis of response frequencies suggests that the characteristic components of the most detrimental Western dietary patterns (e.g., fast-food consumption) are largely absent in the patient cohort, which may be considered a positive finding. At this age, the influence of peers also has a major impact on nutritional behavior and should not be underestimated.

Our findings highlight a gap between health literacy and practical dietary behavior, suggesting that even patients with sufficient HL may benefit from targeted nutritional counseling. Monitoring, regular nutritional counselling with a dietitian and individualized diet also in remission phase are key. It is crucial that the principles of healthy eating, emphasizing the benefits of omega-3 intake through oily fish and nuts and the importance of regular consumption of fruits and vegetables in remission phase are systematically addressed. Nonetheless, patient-centered communication remains a responsibility for health professionals within every session, as it contributes to improve adherence to medication and diet. Health literacy could be enhanced if schools organized health days more frequently and integrated education on health maintenance into the curriculum. Parents should be encouraged and motivated to attend regular consultations with registered dietitians. Moreover, social media is an important source of information for adolescents; providing credible, age-appropriate content through these platforms could also help improve their HL.

Further research involving a larger sample size is required to clarify associations between health literacy and nutritional patterns and to enhance the generalizability of the findings.

### Limitations

Despite our best efforts, the study has its own limitations, especially its low number of patient cases. It is caused by multiple factors starting with the selected target group has a relatively rare disease. Additionally, data collection was limited to patients receiving biological therapy at the clinic, which allowed sufficient time for questionnaire administration. This treatment modality is also typically associated with long-term remission, often enabling patients to follow a “normal diet”. Patients on restricted diets (e.g., EEN, CDED) or with other conditions (e.g., ASD) that affect their diet were excluded, which further narrowed the pool of eligible patients.

Unfortunately, many patients or their caregivers refused to participate as well. Furthermore, for consistent data we only executed the KIDMED questionnaire on patients being in remission phase which means even lower number of cases to report about.

Data on the sex of control participants were not recorded due to logistical limitations during data collection.

Another limitation factor arises from the age characteristics of our sample. The age of the 99 participants overall did not vary considerably, as the majority of participants fell within a relatively narrow age range, with the central 50% belonging to the 15–16-year-old age group. At the same time, the full age range of the sample was substantial: the youngest patient was 10 years old, while the oldest was 19. However, this wide range did not allow for meaningful comparisons between different age groups, as the distribution was strongly skewed towards the 15–16-year-old group.

## 5. Conclusions

In this study, we assessed health literacy and dietary adherence to the Mediterranean diet among adolescent patients with inflammatory bowel disease. Our findings indicate that while the majority of participants demonstrated adequate health literacy, this did not translate into healthier dietary practices, as adherence to the Mediterranean diet was generally poor. Notably, although the most harmful dietary patterns, such as frequent fast-food consumption, were largely absent, protective elements like regular intake of fish, legumes, and nuts were also lacking.

The positive association between health literacy and age aligns with previous research. However, no significant associations were found between health literacy and dietary adherence in this sample, which may partly be attributed to the limited sample size. These results highlight the gap between theoretical health knowledge and practical dietary behavior among adolescents with IBD.

Our findings underscore the need for targeted, personalized nutritional counseling for this patient population, even during remission phases. Future studies with larger and more diverse cohorts are warranted to clarify the relationship between health literacy and dietary patterns and to inform effective interventions aimed at improving long-term health outcomes in this vulnerable group.

## Figures and Tables

**Figure 1 nutrients-17-02458-f001:**
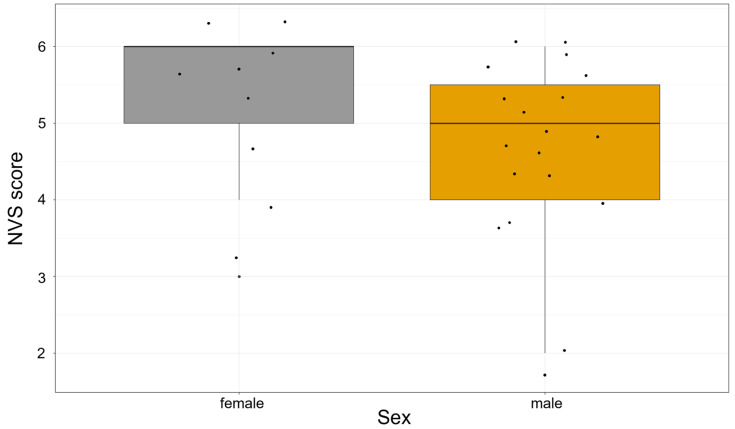
NVS score by sex in the patient group. Boxplot illustrating the distribution of Newest Vital Sign (NVS) health literacy scores by sex. Median scores are indicated by the horizontal lines within the boxes. The interquartile ranges (IQR) are represented by the box boundaries. Individual data points are displayed as dots. No significant difference was observed between male and female participants.

**Figure 2 nutrients-17-02458-f002:**
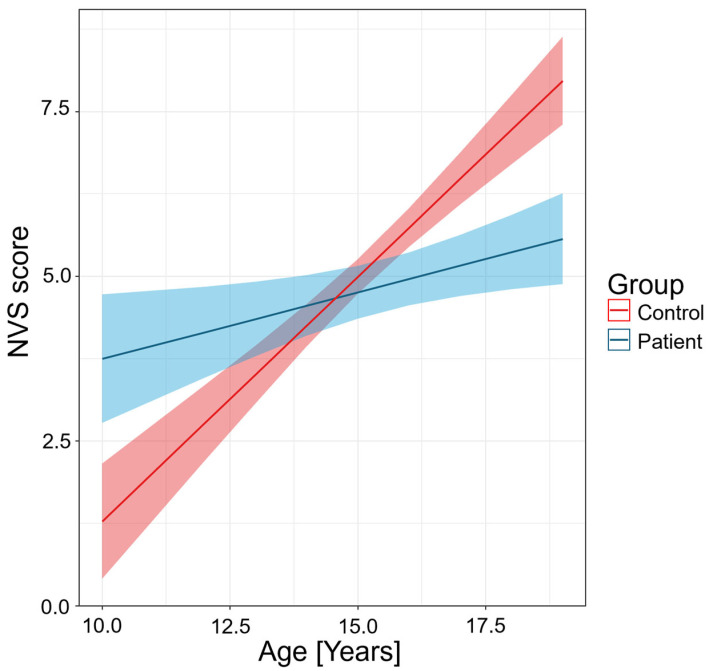
Predicted values of NVS (Newest Vital Sign) score by age and group. The figure shows predicted NVS scores by age for patients (blue) and controls (red), based on linear regression models. NVS scores increase with age in both groups, but the rise is steeper in controls. Shaded areas represent 95% confidence intervals.

**Figure 3 nutrients-17-02458-f003:**
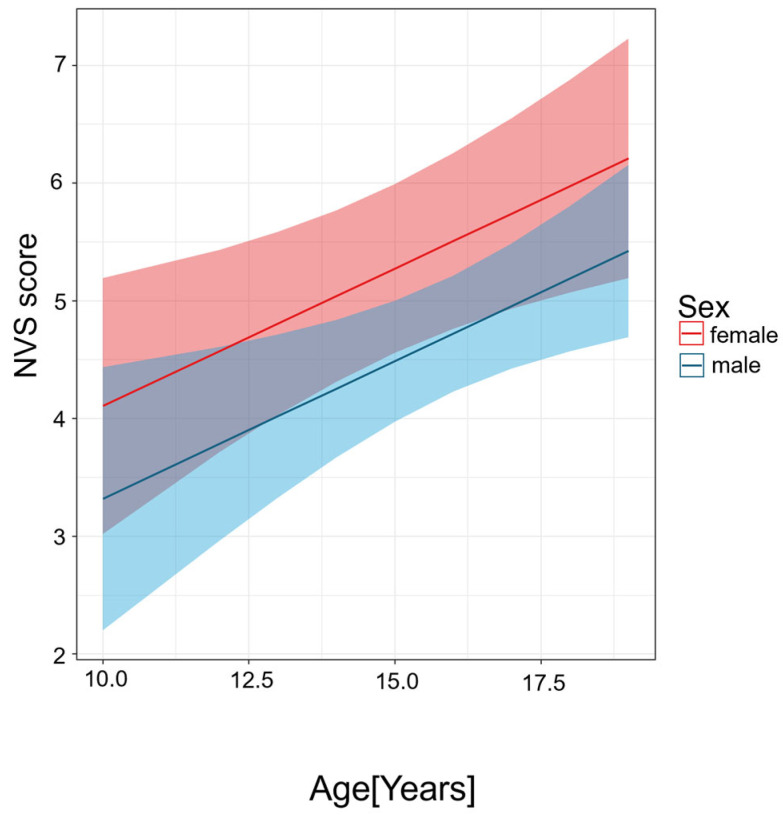
Predicted values of NVS score in patients by age and sex. The figure shows predicted NVS scores by age for male (blue) and female (red) participants, based on linear regression models. NVS scores increase with age in both groups. Shaded areas represent 95% confidence intervals.

**Figure 4 nutrients-17-02458-f004:**
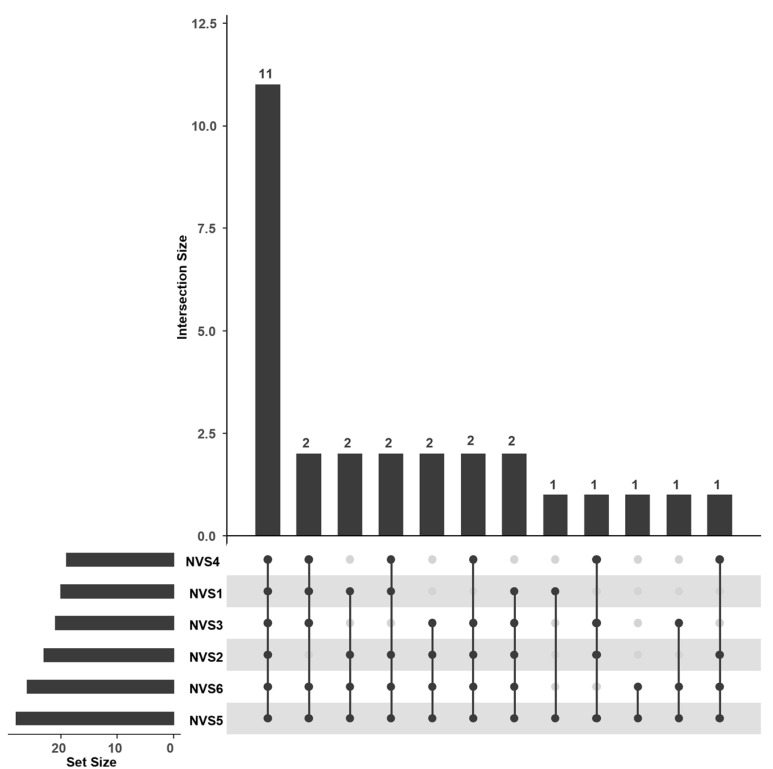
Upset plot for Newest Vital Sign “subscales”. Question 4 (NVS4); Question 1 (NVS1); Question 3 (NVS3); Question 2 (NVS2); Question 6 (NVS6); Question 5 (NVS5). The upset plot represents the response pattern for the Newest Vital sign tool. The dots show variable combinations in the top section. Individual variable frequencies are on the left, and combination frequencies are at the top.

**Figure 5 nutrients-17-02458-f005:**
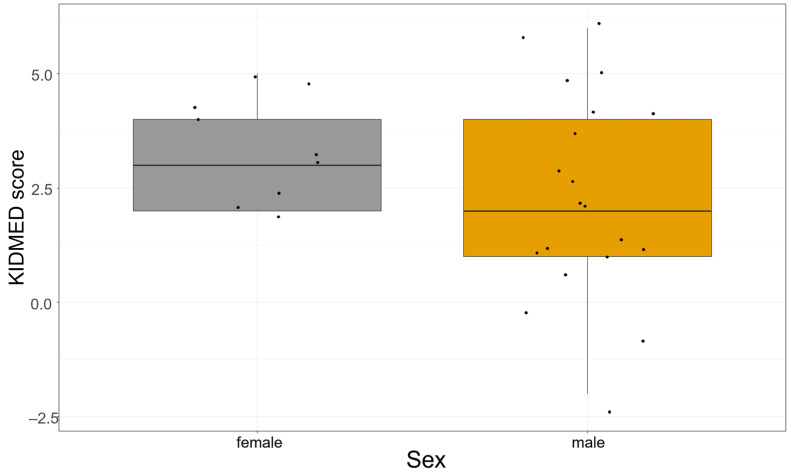
KIDMED score by sex. Boxplot illustrating the distribution of KIDMED dietary adherence scores by sex. Median scores are indicated by the horizontal lines within the boxes. The interquartile ranges (IQR) are represented by the box boundaries. Individual data points are displayed as dots. No significant difference was observed between male and female participants.

**Figure 6 nutrients-17-02458-f006:**
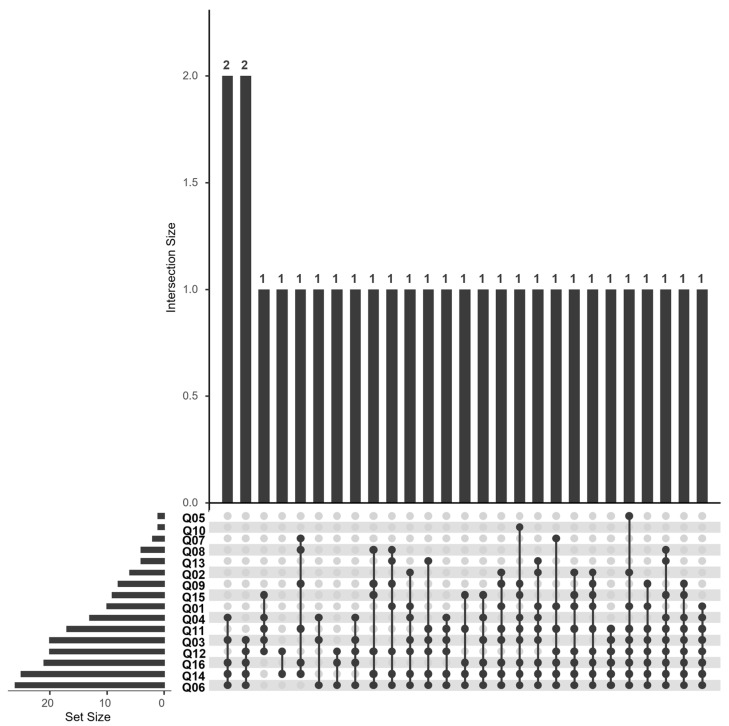
Upset plot KIDMED score “subscales”. Q stands for “Question”, and the following number refers to the item in the KIDMED questionnaire. The upset plot represents the response pattern for the KIDMED Mediterranean diet adherence screening tool. The dots show variable combinations in the top section. Individual variable frequencies are on the left, and combination frequencies are at the top.

**Table 1 nutrients-17-02458-t001:** Descriptive table for control and patients for health literacy.

	Control (N = 63)	Patients (N = 28)	Overall (N = 91)
Age (years)			
Mean	15.2	15.7	15.4
SD ^1^	1.63	2.35	1.93
NVS ^2^ score			
Mean (SD)	5.13 (1.58)	4.86 (1.18)	5.04 (1.47)
Median (Q1 ^3^, Q3 ^4^)	6.00 (5.00, 6.00)	5.00 (4.00, 6.00)	6.00 (4.50, 6.00)
Min ^5^, Max ^6^	0, 6.00	2.00, 6.00	0, 6.00
NVS category			
high likelihood of limited HL ^7^	4 (6.3%)	0 (0%)	4 (4.40%)
possibility of limited HL	4 (6.3%)	3 (10.71%)	7 (7.69%)
adequate HL	55 (87.3%)	25 (89.29%)	80 (87.91%)

Abbreviations: ^1^ SD = Standard deviation; ^2^ NVS = Newest Vital Sign; ^3^ Q1 = Quartile 1; ^4^ Q3 = Quartile 3; ^5^ Min = Minimum; ^6^ Max = Maximum; ^7^ HL = Health literacy.

**Table 2 nutrients-17-02458-t002:** NVS score vs. groups adjusted to age.

Characteristic	Beta	95% CI ^1^	*p*-Value
**Group**			**<0.001**
*Controls*	–	–	
*Patients*	7.9	4.3, 11	
**Age [Years]**	0.74	0.58, 0.90	**<0.001**
**Group * Age [Years]**			**<0.001**
*Patients * Age [Years]*	−0.54	−0.77, −0.31	

Abbreviation: ^1^ CI = Confidence Interval.

**Table 3 nutrients-17-02458-t003:** NVS score vs. sex adjusted to age (patients only).

Characteristic	Beta	95% CI ^1^	*p*-Value
**Sex**			0.080
*Female*	–		
*Male*	−0.79	−1.7, 0.10	
**Age [Years]**	0.23	0.06, 0.41	**<0.010**

Abbreviation: ^1^ CI = Confidence Interval.

**Table 4 nutrients-17-02458-t004:** Descriptive table for diet adherence.

	Female (n = 9)	Male (n = 20)	Overall (N = 29)
Age (years)			
Mean	14.9	16.0	15.7
SD ^1^	2.71	2.12	2.35
KIDMED score			
Mean (SD)	3.33 (1.22)	2.35 (2.25)	2.66 (2.02)
Median (Q1 ^2^, Q3 ^3^)	3.00 (2.00, 4.00)	2.00 (1.00, 4.00)	3.00 (1.00, 4.00)
Min ^4^, Max ^5^	2.00, 5.00	−2.00, 6.00	−2.00, 6.00
KIDMED category			
poor adherence	4 (44.44%)	13 (65.00%)	17 (58.62%)
average adherence	5 (55.56%)	7 (35.00%)	12 (41.38%)
good adherence	0 (0%)	0 (0%)	0 (0%)

Abbreviations: ^1^ SD = Standard deviation; ^2^ Q1 = Quartile 1; ^3^ Q3 = Quartile 3; ^4^ Min = Minimum; ^5^ Max = Maximum.

## Data Availability

Records and data pertaining to this study are stored electronically at the Semmelweis University, Faculty of Health Sciences, Department of Dietetics and Nutritional Sciences, Hungary, and can be provided by the corresponding author on a reasonable request.

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
