# Peer review of "Health Literacy and Nutrition of Adolescent Patients with Inflammatory Bowel Disease"

_nutrients, 2025, doi:10.3390/nu17152458_

Round 1
Reviewer 1 Report
Comments and Suggestions for Authors
In this manuscript, authors have tried to assess health literacy (HL) and dietary patterns in adolescent and pediatric patients with IBD. Seen from a global perspective, there are not many papers about this relationship, so medical experts would have high expectations regarding the findings. However, there are several problems with the manuscript that make it difficult to publish as is. I think the following points should be improved.
1) In Introduction, Page 1 line 37, it is highly disconcerting that the introduction begins with “However”.
2) In Introduction, Page 2 line 72-76, these sentences are not needed in this section.
3) The background on measurement tools such as NVS and KIDMED should be explained in the Introduction, not Material&Methods.
3) In Materials and Methods, there are no headings, and it is very difficult to understand what you want to explain. At a minimum, there should be the headings “Ethics Statement,” “Subjects (Participants),” “Methods,” and "Statistical Analysis. The contents should be described according to each heading. On the contrary, the heading “Limitations” is not appropriate for the “Materials and Methods” section. Such contents should be mentioned at the “Discussion” section.
4) In Figure 1-3 and 5, texts and numerical values in the graphs are too small and should be displayed in a larger size. In addition, the numbers and letters overlap in some places and are extremely close to each other, which should be improved.
5) The titles of the Figures are present, but there is no description (annotation) following the titles. Therefore, each Figure is uninformative to the reader.
6) The KIDMED score was assessed only in the patient group (Table 4 and Figure 5), making it impossible to compare diet quality with the control group. As a result, it is unclear whether poor diet adherence is due to IBD or other demographic factors. Please include KIDMED scores for the control group to enable group comparisons.
7) Although patients with Crohn’s disease and ulcerative colitis are included in IBD, subgroup analysis by disease type is absent, which may obscure disease-specific patterns. Please add disease-specific analyses to explore differences between Crohn’s disease and ulcerative colitis patients.
8) There are multiple problems with the discussion part. First, there is not enough text in the discussion. While prior studies are cited, the discussion does not adequately compare or contrast this study’s findings with existing literature, weakening its contribution to the field. The limitations related to the single-center design and inclusion of only patients in remission are not sufficiently emphasized in the discussion. While the need for nutritional counseling is mentioned, there is a lack of discussion about specific intervention strategies or how HL could be improved in practice. While, the authors overemphasize trends (e.g., higher scores in females) despite a lack of statistical significance. This could mislead readers into interpreting non-significant findings as meaningful. Please avoid overstating non-significant findings; describe them clearly as trends if necessary.

Author Response
Comment 1: In Introduction, Page 1 line 37, it is highly disconcerting that the introduction begins with “However”.
Response 1: Thank you for pointing this out. Therefore we have rephrased the first sentence on page 1, paragraph 1, lines 38-39. „Inflammatory bowel disease (IBD) profoundly affects patients’ nutrition; however no universally recommended diet exists that would suit all patients”
Comment 2: In Introduction, Page 2 line 72-76, these sentences are not needed in this section.
Response 2: Thank you for your comment. We prepared the manuscript using the MDPI “Instructions for Authors” (https://www.mdpi.com/journal/nutrients/instructions), which specify in the “Introduction” section: “Finally, briefly mention the main aim of the work and highlight the principal conclusions”. Therefore, we decided to leave this paragraph unchanged.
Comment 3: The background on measurement tools such as NVS and KIDMED should be explained in the Introduction, not Material&Methods.
Response 3: We partly agree with this comment. We have moved the description of the tools to the Introduction, while the detailed explanation of why these tools were used has been kept in the “Materials and Methods” section. Changes can be found on page2, lines 55-59 and lines 75-78.
Comment 4: In Materials and Methods, there are no headings, and it is very difficult to understand what you want to explain. At a minimum, there should be the headings “Ethics Statement,” “Subjects (Participants),” “Methods,” and "Statistical Analysis. The contents should be described according to each heading. On the contrary, the heading “Limitations” is not appropriate for the “Materials and Methods” section. Such contents should be mentioned at the “Discussion” section.
Response 4: Thank you for pointing this out. Therefore we added subheadings (Study Design and Participants, Data Collection Period and Ethics Statement, Methods, Participants’ Characteristics, Statistical Analysis) to this section and moved “Limitations” to the “Discussion” section. Changes can be found on page 2, paragraph 1, line 86; paragraph 2, line 92; paragraph 3, line 96; page 3, paragraph 4, line 109; paragraph 5, line 118; page 11, paragraph 5, lines 373-393.
Comment 5: In Figure 1-3 and 5, texts and numerical values in the graphs are too small and should be displayed in a larger size. In addition, the numbers and letters overlap in some places and are extremely close to each other, which should be improved.
Response 5: Agree. We have accordingly edited and corrected the figures to a larger size.
Comment 6: The titles of the Figures are present, but there is no description (annotation) following the titles. Therefore, each Figure is uninformative to the reader.
Response 6: Thank you for pointing this out. We added description to each figures accordingly. The annotations can be found on page 4, lines 157-161; page 5 lines 177-180; page 6 lines 196-198; page 7, lines 207-211; page 9, lines 250-253; page 9, lines 268-272.
Comment 7: The KIDMED score was assessed only in the patient group (Table 4 and Figure 5), making it impossible to compare diet quality with the control group. As a result, it is unclear whether poor diet adherence is due to IBD or other demographic factors. Please include KIDMED scores for the control group to enable group comparisons.
Response 7: Thank you for this valuable comment. We fully agree that including KIDMED scores for the control group would allow for a more thorough comparison between groups. However, as we clarified on page 8, line 232, the KIDMED questionnaire was only administered to patients with IBD in this study. Therefore, unfortunately, we do not have dietary data available for the control group.
Comment 8: Although patients with Crohn’s disease and ulcerative colitis are included in IBD, subgroup analysis by disease type is absent, which may obscure disease-specific patterns. Please add disease-specific analyses to explore differences between Crohn’s disease and ulcerative colitis patients.
Response 8: Thank you for this comment. We added the subgroup analyses, updated text can be found on page 6, paragraph 2, lines 201-203 “For the analysis of the results by disease type, the Mann-Whitney U test was per-formed; however, no significant difference was found in the NVS scores between CD and UC patients (U = 65, z = 0.028, p = .976).”and page 9, paragraph 2, lines 256-257 “The Mann-Whitney U Test detected no statistically significant differences between the KIDMED score results of CD and UC patients (U = 62, z = 0.349, p = .726).”
Comment 9: There are multiple problems with the discussion part. First, there is not enough text in the discussion. While prior studies are cited, the discussion does not adequately compare or contrast this study’s findings with existing literature, weakening its contribution to the field. The limitations related to the single-center design and inclusion of only patients in remission are not sufficiently emphasized in the discussion. While the need for nutritional counseling is mentioned, there is a lack of discussion about specific intervention strategies or how HL could be improved in practice. While, the authors overemphasize trends (e.g., higher scores in females) despite a lack of statistical significance. This could mislead readers into interpreting non-significant findings as meaningful. Please avoid overstating non-significant findings; describe them clearly as trends if necessary.
Response 9: Thank you for your comment. We extended the “Discussion” section with detailed comparisons with the literature and mentioned intervention strategies for improving HL. Changes can be found on page 10, paragraph 9, lines 307-313 “In patients with IBD, most health literacy studies focus specifically on disease-related knowledge and consistently report low levels of health literacy. Limited HL has been linked to worse patient-reported outcomes, depressive symptoms, and poorer healthcare outcomes—including longer hospital stays, higher readmission and complication rates—and some estimates suggest that up to 24% of IBD patients have inadequate health literacy [22,24]. Similar trends are observed in general gastrointestinal cohorts, where lower health literacy predicts increased hospital length of stay, readmissions, and com-plications [37].”; page 11, lines 328-329 “Similarly, as age increased, the HL score also increased proportionally.”; page 11, paragraph 1, lines 335-347 “Fiorindi et al. reported that patients demonstrated good dietary adherence, as indicated by an average Medi-Lite score of 10.4 [13]. However, they similarly found no significant dif-ferences between the CD and UC groups, as our study. Dietary adherence appeared to be more strongly associated with disease activity, with patients with CD showing signifi-cantly higher adherence during remission. This aspect was not investigated in the present study, as it was assumed a priori that patients would not follow a balanced diet during periods of disease flare. Based on Strisciuglio’s findings, who utilized the same dietary adherence measurement tool as our research, children with IBD had an intermediate ad-herence to MD [12], and they also were able to detect a correlation about MD and de-creased intestinal inflammation. Cadoni et al. reported about a low adherence to MD among IBD patients, similarly to our study, as measured by the Medi-Lite score [14]. Our observation of low dietary adherence among patients in remission may be due to several underlying factors, the examination of which is recommended.”; page 11, paragraph 3, lines 364-369 “Health literacy could be enhanced if schools organized health days more frequently and integrated education on health maintenance into the curriculum. Parents should be encouraged and motivated to attend regular consultations with registered dietitians. Moreover, social media is an important source of information for adolescents; providing credible, age-appropriate content through these platforms could also help improve their HL.”
We rephrased and updated the text to avoid overstating non-significant findings according to your advise and added the results of statistical analyses to clarify the findings. Updated text is on page 1, paragraph 1, line 28 “Most participants (87.9%) had an adequate HL, which was positively associated with age.”; page 3, paragraph 10, lines 140-142 “The mean score was slightly lower in the patient group (4.86 ± 1.18) vs. in the control group (5.13 ± 1.58); however, this difference was not statistically significant (Mann-Whitney U = 660.5, z = 1.900, p = .057).”; page 4, paragraph 1, lines 152-155 “It should be noted that this difference did not reach statistical significance based on the Mann-Whitney U test (U = 61; the critical value of U at p < .05 is 45). Accordingly, the result was not significant at the p < .05 level (z = -1.181, p = .238).”; page 8, paragraph 2, lines 238-240 “Although the mean score was higher among females (3.33 ± 1.22) compared to males (2.35 ± 2.25), the difference was not statistically significant (Mann-Whitney U = 60.5, z = -1,367, p = .171).” and lines 240-241 “Among female adolescents the minimum score was 2 and the median 3, among males the minimum was -2 and the median 2.”, page 10, paragraph 4, lines 288-289 “Reading comprehension skills were found to be better than percentage numeracy and HL is expected to increase with age.”
Reviewer 2 Report
Comments and Suggestions for Authors
Dear Authors, I found your study interesting, but some changes should be done to improve this manuscript.
In abstract and methodology section it should be added what stage (remission or exacerbation) of the disease were the patients in? It is written in the limitations (lines 128-131 and 134) but this is important for the characteristics of the study group and it should be written in the methodology section and shortly in abstract.
I suggest to move limitation section after the discussion and before conclusion section.
In the introduction section lines 72-76 are rather conclusions and should be moved to the conclusion section.
The NVS questions should be written in the methodology.
All abbreviations from Table 1 should be explained below the table.
In results (point 3.1.) the authors wrote that “The mean score was slightly lower in the patient group (4.86 ± 1.18) vs. in the 153 control group (5.13 ± 1.58)”, and further, that the “mean 161 NVS score was higher in females (5.22) compared to males (4.68)”. Statistical analysis should be done to compare these values, and the results should be shown.
All abbreviations from Table 3 and 4 should be explained below these tables.
It would be good to add conclusion section after the discussion.
Author Response
Comment 1: In abstract and methodology section it should be added what stage (remission or exacerbation) of the disease were the patients in? It is written in the limitations (lines 128-131 and 134) but this is important for the characteristics of the study group and it should be written in the methodology section and shortly in abstract.
Response 1: Agree. The updated text can be found on page 1, paragraph 1, lines 24-26 “HL was assessed using the Newest Vital Sign (NVS) tool regardless of disease activity, whereas diet quality was evaluated by the KIDMED questionnaire exclusively in patients in remission.”; page 3, paragraph 4, lines 113-115 “The KIDMED questionnaire was administered only to patients in remission to avoid distortion of the results due to dietary changes during disease flare-ups.”
Comment 2: I suggest to move limitation section after the discussion and before conclusion section.
Response 2: Thank you for pointing this out. We have moved “Limitations” to the “Discussion” section. Updated text can be found on page 11, paragraph 5, lines 373-393.
Comment 3: In the introduction section lines 72-76 are rather conclusions and should be moved to the conclusion section.
Response 3: Thank you for your comment. We prepared the manuscript using the MDPI “Instructions for Authors” (https://www.mdpi.com/journal/nutrients/instructions), which specify in the “Introduction” section: “Finally, briefly mention the main aim of the work and highlight the principal conclusions”. Therefore, we decided to leave this paragraph unchanged.
Comment 4: The NVS questions should be written in the methodology.
Response 4: Thank you for your comment. We decided to leave the two NVS questions in this section in order to facilitate the reader’s understanding of the content.
Comment 5: All abbreviations from Table 1 should be explained below the table.
Response 5: We agree with this comment. Therefore, we added abbreviations to each table. Updated text is on page 4, lines 148-149.
Comment 6: In results (point 3.1.) the authors wrote that “The mean score was slightly lower in the patient group (4.86 ± 1.18) vs. in the 153 control group (5.13 ± 1.58)”, and further, that the “mean 161 NVS score was higher in females (5.22) compared to males (4.68)”. Statistical analysis should be done to compare these values, and the results should be shown.
Response 6: Agree. We have added the results of statistical analyses to clarify the findings. Changes can be found on page 3, paragraph 10, lines 140-142 “The mean score was slightly lower in the patient group (4.86 ± 1.18) vs. in the control group (5.13 ± 1.58); however, this difference was not statistically significant (Mann-Whitney U = 660.5, z = 1.900, p = .057).”; page 4, paragraph 1, lines 151-155 “Sex data were not available for the control group; however, among patients, the mean NVS score was higher in females (5.22) compared to males (4.68). It should be noted that this difference did not reach statistical significance based on the Mann-Whitney U test (U = 61; the critical value of U at p < .05 is 45). Accordingly, the result was not significant at the p < .05 level (z = -1.181, p = .238).”
Comment 7: All abbreviations from Table 3 and 4 should be explained below these tables.
Response 7: We agree with this comment. Therefore, we added abbreviations to each table. Updated text is on page 6, line 192; page 8, lines 244-245.
Comments 8: It would be good to add conclusion section after the discussion.
Response 8: Thank you for this comment. In accordance with the request, we have revised the manuscript to include “Conclusion” section. It can be found on page 12, lines 394-411.
Round 2
Reviewer 1 Report
Comments and Suggestions for Authors
The authors have carefully addressed the reviewers' comments and made thoughtful revisions throughout the manuscript. Their responses demonstrate a clear understanding of the concerns raised, and the changes have significantly improved the clarity and scientific quality of the work.